# Experimental Study on Seismic Behavior of Newly Assembled Concrete Beam–Column Joints with L-Shaped Steel Bars

## Mengjiao Lv, Taochun Yang * and Mingqiang Lin

College of Civil Engineering and Architecture, University of Jinan, Jinan 250022, China;
lvmengjiao2021@126.com (M.L.); cea_linmq@ujn.edu.cn (M.L.)
* Correspondence: cea_yangtc@ujn.edu.cn

**Abstract:** A novel concrete beam–column connection utilizing L-shaped steel bars is proposed to address the growing demand for prefabricated buildings and to ensure good seismic performance in such beam–column structures. After positioning two prefabricated beams with L-shaped tendons into the designated connection points at the top and bottom of the columns, concrete is poured into the post-cast section of the joint and the composite beam area, realizing a connection between the beams and columns. Quasi-static tests were performed on four combined backbone curves and one cast-in-place joint to investigate their failure modes and stress mechanisms. Through low-cycle repeated loading tests, it is found that measures such as increasing the area of the post-cast concrete in the joint area, the length of the L-shape, and the concrete strength in the composite beam area can effectively improve the bonding ability between the post-cast area of the joint specimens and the precast members, to improve the ductility performance, energy dissipation capacity, and bearing capacity of the joint specimens. The initial stiffness of the joint can be effectively improved by presetting the steel pipe in the column. Concurrently, the finite element method (FEM) was employed for parameter analysis. By integrating the test and FEM results, an equation for calculating the shear capacity of the connection was derived. The findings demonstrate that the hysteresis curve of the newly assembled joints is full, and its overall performance index is roughly the same as that of the cast-in-place joints. Additionally, enhancing the post-casting area of concrete, the length of the L-shaped bars, the concrete strength in the composite beam region, the axial compression ratio, or the steel tube dimensions can effectively improve the overall performance. The derived equation for the shear-bearing capacity of the connection satisfies design and application requirements.

**Keywords:** assembled beam–column joints; experimental study; seismic performance; shear bearing capacity





## 1. Introduction

Since the 1980s, prefabricated concrete structures have gained widespread adoption worldwide. As early as the 1970s, the United States began investigating prefabricated concrete structures, with the American Concrete Institute's ACI 318-11 "Building Code Requirements for Structural Concrete" and the "PCI Design Manual" making significant contributions to the development of prestressed concrete. The European prefabricated concrete building industry boasts a long history that has shaped the industrialization of these structures and yielded a relatively comprehensive standard system. The latest research on the performance of prefabricated concrete structures is summarized and published in "Code and Standard 2010". Japan has rigorously explored the seismic performance of prefabricated structures and provided a comprehensive explanation of the standards for prefabricated concrete building systems spanning design, manufacturing, and construction. In September 2016, China's General Office of the State Council issued the "Guiding Opinions on Vigorously Developing assembled Buildings", proposing that, within 10 years, prefabricated buildings should account for 30% of new constructions [1–4]. Numerous earthquake

disaster investigations revealed that damages to prefabricated structural joints were the primary cause of building damage, significantly impacting seismic performance [5–8]. Consequently, researchers worldwide have conducted extensive investigations on the joints of prefabricated concrete structures, encompassing steel bar anchorage connections, grouting sleeve connections, welding, and bolt connection [9–12].

Ketiyot [13] introduced a novel T-section steel structure connection joint, incorporating an embedded T-section steel and an embedded steel plate welded to the precast beam. The study indicates an enhancement in the seismic resistance of this connection; however, there is a rapid degradation in stiffness, and the bearing capacity is lower than anticipated. Yuskel [14] presented a post-cast concrete joint with a sleeve that includes a grouting slot in the upper portion of the column end, which is filled after the insertion of the steel bar into the assembled steel sleeve. The research suggests reduced damage and improved ductility of this joint compared to the cast-in-place joint, with a minor effect on the bearing capacity. Yang Hui [15] devised a novel style of prestressed hybrid beam–column joint, employing the post-tensioned prestressed methodology for connecting the column and beam, casting the remainder of the joint in place. The research found that the ductility of the novel joint has improved, with minimal change in the bearing capacity. Loo and Yao [16] designed 18 half-scale concrete frame joints, with testing results establishing that the post-cast integral beam–column joints' strength, seismic performance, and ductility surpass those of the cast-in-site joints. Englekirk [17] recommended a precast concrete frame joint featuring a ductile connector, with testing indicating commendable energy dissipation and recovery characteristics. Exploring the seismic performance of assembled steel-reinforced concrete beam–column joints, Li [18] developed two groups of two full-scale joint specimens, providing evidence of exceptional ductility and seismic performance in both groups. Lai [19] introduced a novel joint composed of a fan-shaped lead viscoelastic damper and an assembled concrete frame, with experimentation proving a solid energy dissipation effect, along with substantial improvements in the joint's bearing capacity and displacement ductility. Gu [20] suggested a new variety of hooked assembled concrete beam–column joint, with testing confirming excellent energy dissipation and ductility. Ji [21] introduced a concrete-filled steel tubular column joint with only ring bars and no ring beams, showing that the hysteresis loop of the joint is quite full, and it possesses considerable energy consumption capacity and ductility. Miao [22] suggested a novel connection using high-strength bolts to join the assembled beam and assembled column through the embedded steel section, with testing indicating that the seismic performance of the new assembly frame connection parallels that of the traditional cast-in-site joint.

Through the aforementioned research, it is evident that, while the general performance of steel bar anchorage connection is enhanced, the quality assurance for concrete pouring is challenging. Grouting sleeve connection technology is comparatively mature, yet the quality of the sleeve connection requires verification, and while the welding bolt connection and installation process is convenient and time-efficient, it demands high precision. The prestressed connection greatly enhances the stiffness, strength, and restoring force of joints, but its energy dissipation capacity is poor. At present, the most mature and widely used technology is wet connection. Although the overall performance and seismic performance reach or even exceed the cast-in-place joints, the longitudinal beams and columns in the core area of the joints are mainly connected using sleeve grouting and welding, and the construction is complicated, and the quality of the joints is not easy to guarantee. At the same time, due to the large size of the prefabricated components, it is not conducive to the transportation of components, which also limits the promotion and application of such connection methods.

Aiming at the common problems of prefabricated joints, a novel type of assembled concrete beam–column L-shaped reinforced connection joint is proposed in this paper. The joint has the advantages of a simple structure and good economy, which can ensure the overall stiffness of the beam–column joint in the process of transportation and installation, simple construction and effective construction quality. To study the seismic performance

of the newly assembled joints, four combined joints, and one cast-in-place joint were designed and fabricated. The pseudo-static test was carried out on them. The corresponding seismic performance curves were obtained through comparative analyses of the test results, and their failure modes and application mechanisms were studied. At the same time, the finite element method was used to analyze the parameters of the simulation results and the test results, and the formula for calculating the shear strength of the L-shaped joint of the novel assembled concrete beam–column was obtained. This can meet the growing demand for assembled buildings and ensure good seismic performance in such beam–column structures.

## 2. Joint Structure

The L-shaped reinforced connection joints primarily connect beams and columns by pouring concrete into the post-cast section of the joints and the composite beam area. This process occurs after two precast beams with L-shaped tendons have been installed on the precast column and the designated position of the lower node area. The upper and lower parts of the precast concrete columns are fixed on the surface using round steel tubes with shear bolts, which serve to increase the bonding force between the concrete and round steel tubes. The new L-shaped reinforced joints are constructed by post-pouring in one piece, and they are poured twice. In contrast, the conventional construction method involves one-time pouring for precast beams and columns. The key difference lies in the separation of steel bars for beams and columns and the separate support for formwork. The longitudinal bars at the bottom of precast beams are bent upwards by 90 degrees. Stirrups are not initially bound in the core area of precast column joints, with assembly taking place in the factory. After curing, the precast concrete beam will be installed at the designated position in the node area. U-shaped stirrups are then tied in the core area of the column node. Finally, secondary pouring is conducted in the post-pouring area of the concrete beam–column node and the cast-in-place area at the upper part of the composite beam, achieving a connection between the beams and columns. The joint boasts advantages such as simplicity, robust performance, impressive overall stiffness, easy quality assurance, and convenient transportation of precast components. The joint structure is depicted in Figure 1.

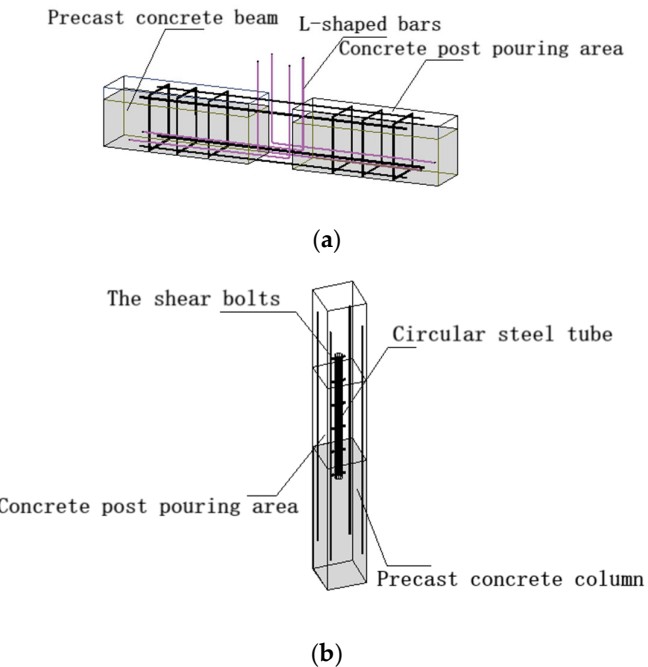

(a)

(b)

**Figure 1.** *Cont.*

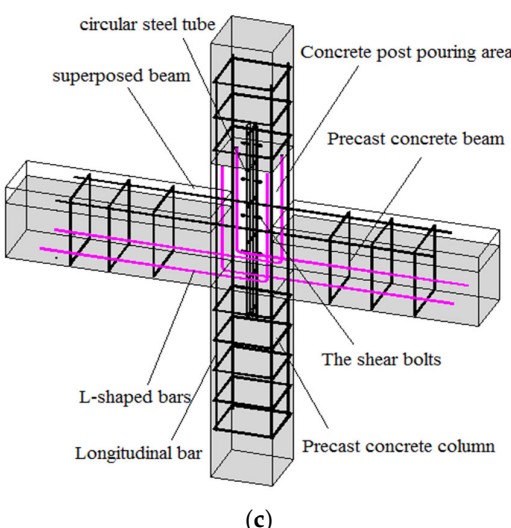

(**c**)

**Figure 1.** Joint structure diagram: (**a**) details of precast beams, (**b**) details of precast column, (**c**) details of joint structure.

## 3. Test

### 3.1. Specimen Design

Mid-column joints in the middle story, which more readily meet simulated boundary conditions, are selected as specimens. The dimensions of the beams and columns are determined according to the position of the reverse bending point when the frame structure supports the horizontal load. Simultaneously, considering the actual range of the laboratory loading device, the design will adhere to a 1:2 scale.

To evaluate the seismic performance of the joint, five joint specimens were fabricated for the test, including one cast in situ joint specimen XJ-1 and four newly assembled joint specimens ZP-1, ZP-2, ZP-3, and ZP-4. Except for the bending length of the L-bar, the concrete strength of the composite slab, and the post-cast concrete area, the other factors for the four assembled specimens are identical. As per Table 1, the difference between ZP-1 and ZP-2 lies in the bending length of the L-bar, which measures 480 mm and 240 mm for these respective samples. The difference between ZP-1 and ZP-3 pertains to the concrete grades of laminated slabs (C30 and C20, respectively). Conversely, the difference between ZP-1 and ZP-4 concerns the area of post-cast concrete. The specific size and reinforcement of the specimen are displayed in Table 1 and Figure 2.

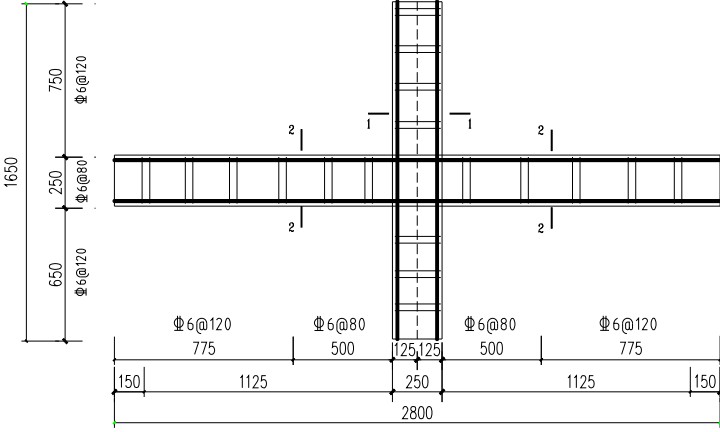

**Figure 2.** *Cont.*

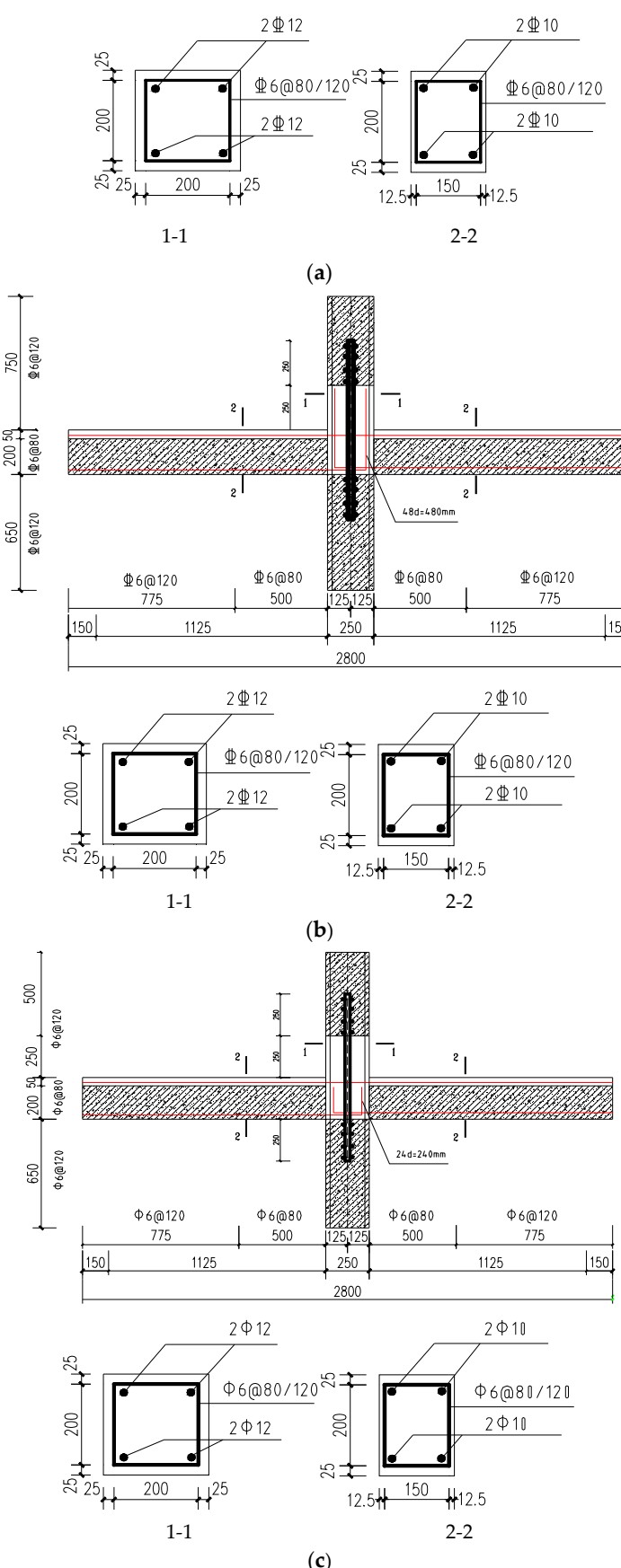

**Figure 2.** *Cont.*

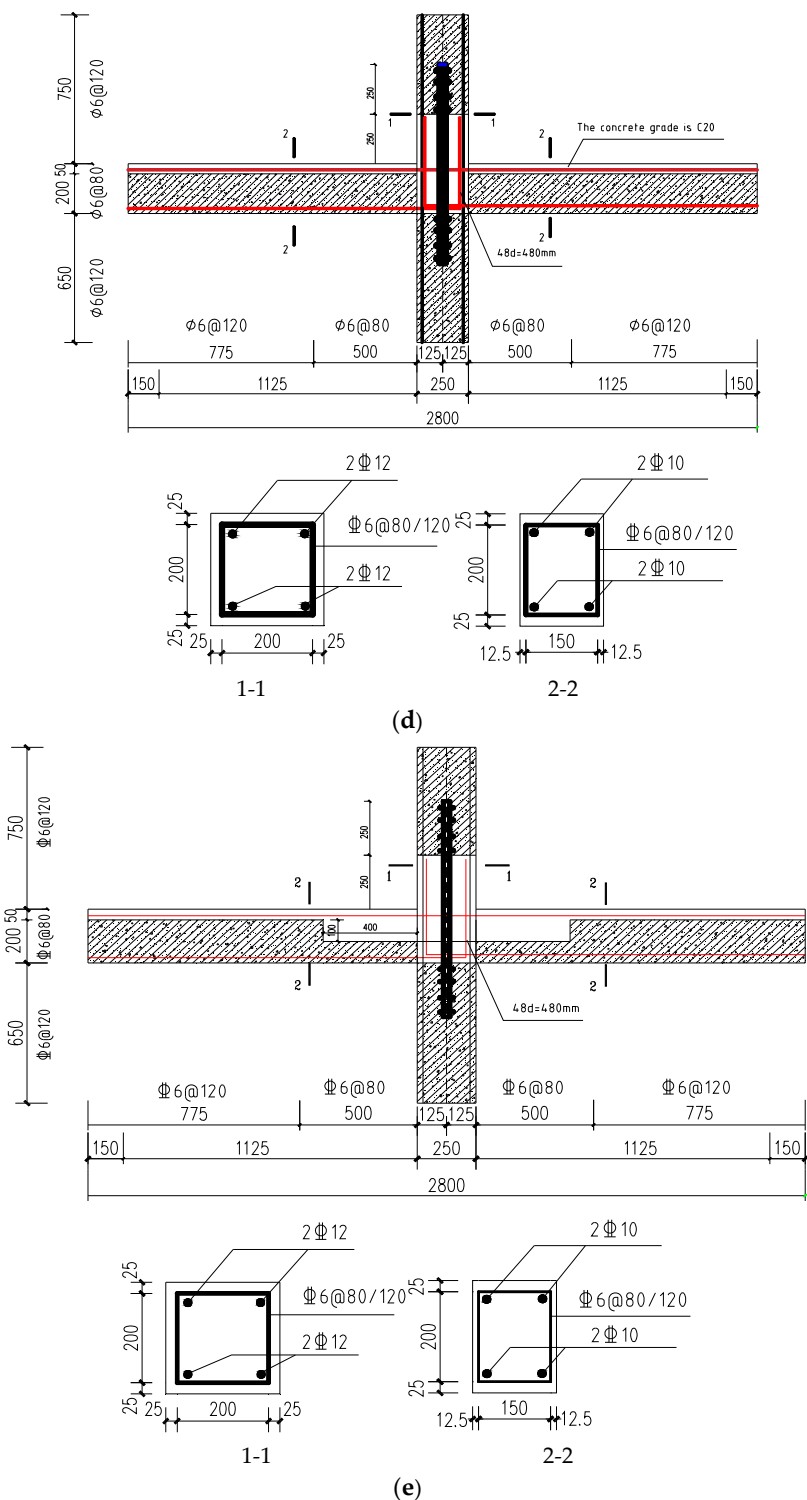

**Figure 2.** Specimen size and reinforcement: (**a**) XJ-1, (**b**) ZP-1, (**c**) ZP-2, (**d**) ZP-3, (**e**) ZP-4. Referring to the Chinese concrete design code, "@" indicates the steel bar spacing and "Φ "indicates the steel bar grade.

**Table 1.** Dimensions and differences of specimens.

| Specimen Number | XJ-1 | ZP-1 | ZP-2 | ZP-3 | ZP-4 |
|---|---|---|---|---|---|
| Section dimensions of beams and columns | Beam: 175 mm × 250 mm; column: 250 mm × 250 mm | | | | |
| Thickness of cover | 25 mm | | | | |
| Dimensions of steel pipe and shear bolts | Steel pipe: d = 42 mm, t = 3 mm, L = 1000 mm; The shear bolts: M12; L = 60 mm | | | | |
| Column stirrups, longitudinal reinforcement | Stirrup: D6@80/120, Longitudinal reinforcement: D12 | | | | |
| Beam stirrups, longitudinal bars | Stirrup: D6@80/120, Longitudinal reinforcement: D10 | | | | |
| Lower longitudinal reinforcement of the beam | Beam longitudinal reinforcement | Lower beam longitudinal bars bend upward 480 mm | Lower beam longitudinal bars bent upward 240 mm | Lower beam longitudinal bars bend upward 480 mm | Lower beam longitudinal bars bend upward 480 mm |
| Length of longitudinal reinforcement at the lower part of the beam | 2750 mm | 1945 mm | 1705 mm | 1945 mm | 1945 mm |
| Concrete strength grade | C30 | C30 | C30 | C20 | C30 |

### 3.2. Material

During the test, reserved steel bars and concrete cube test blocks underwent mechanical property tests in accordance with the Tensile Test of Metallic Materials at Room Temperature (GB/T 228-2010) [23] and the national standard for Testing Methods of Mechanical Properties of Ordinary Concrete (GB/T 50081-2019) [24]. The results are presented in Tables 2 and 3.

**Table 2.** Mechanical properties of reinforcement.

| Steel Type | The Diameter of Steel Pipe (mm) | $f_y$ (MPa) | $f_u$ (MPa) | $E_s$ (GPa) |
|---|---|---|---|---|
| HRB400 | D6 | 429.32 | 540.24 | 200 |
| HRB400 | D10 | 443.25 | 625.16 | 200 |
| HRB400 | D12 | 450.13 | 31.02 | 200 |

where $f_y$ is yield strength, $f_u$ is ultimate strength, $E_s$ is modulus of compressibility.

**Table 3.** Mechanical properties of concrete.

| Specimen | Strength Grade | $f_{cu}$ (MPa) | $f_c$ (MPa) | $E_c$ (MPa) |
|---|---|---|---|---|
| Group A | C30 | 32.61 | 21.81 | $3.65 \times 10^4$ |
| Group B | C30 | 32.72 | 21.88 | $3.66 \times 10^4$ |
| Group C | C20 | 21.81 | 14.59 | $2.66 \times 10^4$ |

where $f_{cu}$ is cube compressive strength, $f_c$ is axial compressive strength, $E_c$ is Modulus of elasticity (Group A is cast in situ joint concrete and precast joint concrete test blocks, Group B is post-cast joint concrete test blocks, and Group C is ZP-3 composite beam concrete test blocks).

### 3.3. Loading Plan

The test utilizes the column end loading mode of beam–column joints. Accommodating the p-Δ effect within the structure under lateral force, the framework's upper column anti-bending points are rendered as moving horizontal hinges when subjected to horizontal load. The bottom column anti-bending points, however, are considered as fixed hinges. The surrounding beams of the joints function as moving horizontal hinges. This complies with the actual stress state. Turn to Figure 3 for the depiction of the loading device.

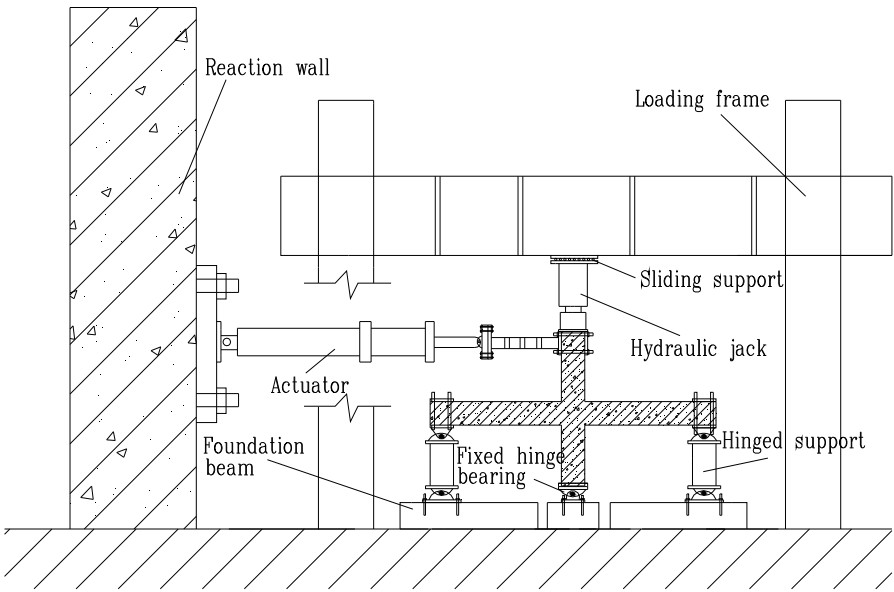

**Figure 3.** Loading device.

Testing loads are segmented into two categories: vertical and horizontal loads. The process commences with a hydraulic jack pushing a steady vertical load of 180 kN at the column peak, considering an axial compression ratio of 0.2 due to limited school laboratory resources. The process is followed by an electro-hydraulic servo actuator (MTS) enforcing cyclic horizontal loading at the upper column peak. To maintain loading steadiness, displacement controls the entire loading. Initial loadings elevate the displacement by 1 mm per stage until load displacement is achieved at 5 mm. Subsequently, each cycle stage undergoes a displacement upsurge of 5 mm three times. Once the load displacement touches 40 mm, the incremental displacement is adjusted to 10 mm, thrice per horizontal cycle. If the load sinks to approximately 85% of the specimens' peak load, the specimen can be considered damaged, and loading is terminated. Figure 4 illustrates the specific loading regime.

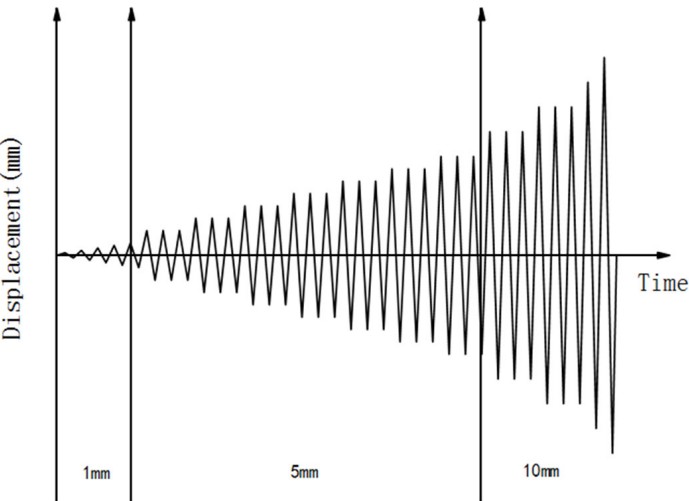

**Figure 4.** Loading regime.

*3.4. Measurement Plan*

The measurement procedures [25] include displacement of the upper column end, support reaction force at the beam end, shear deformation at the joints' core area, rotation angle of the plastic hinge area at the beam–column, overall displacement of the specimen,

and strain and crack width of the reinforced concrete at the pivotal position. A detailed representation of the measuring-gauge placements within and upon the joints can be found in Figure 5.

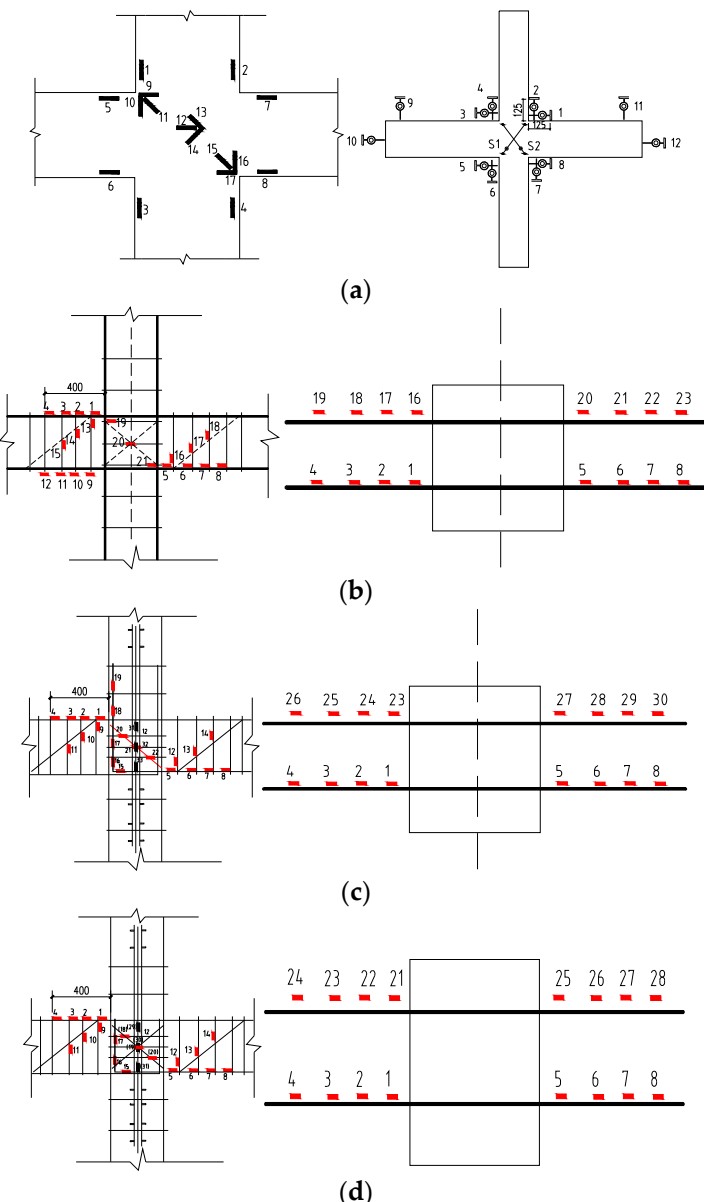

**Figure 5.** Arrangement of strain gauge and displacement gauge: (**a**) arrangement of concrete strain gage and displacement gauge, (**b**) arrangement of strain gauge at cast-in-site joints, (**c**) arrangement of ZP-1, ZP-3 and ZP-4 strain gauges, (**d**) arrangement of ZP-2 strain gauge.

## 4. Phenomenological and Statistical Analysis of Results

### 4.1. Analysis of Test Phenomena

The observation of fissure development during the experimental load application, allied with the quantification of strain and displacement at crucial points, enabled an accurate analysis of joint failure modes. A semi-uniform failure process was noted across the quintet of joint specimens. Initially, single-cycle loads subject to minor amplitude did not visibly alter the concrete surface. However, sequential cyclical loading prompted the formation of vertical fissures around the beam ends. The persistent increase of load displacement instigated appearances of diagonal shear-inclined fractures. The beam-end vertical cracks developed into "U," "Y," and circular fractures over time. When displacement varied

between Δ = ±70 mm~80 mm, the concrete surface started to deteriorate, showcasing a growing number of cracks and significant concrete chunks detaching during loading. This stage also signified a declining bearing capacity of the specimen. When the bearing capacity reduced to around 85% of the maximum load, the specimen was deemed to have suffered damage and the loading was stopped. Specific damages identified included a severe shear failure leading to a significant loss of wedge concrete in ZP-2, attributed to its shorter L-shaped bar. Consequently, ZP-2 was classified as a shear failure. The remaining four joints displayed failures at the beam–column intersection, with the failure mode classified as beam-end failure, as illustrated in Figure 6.

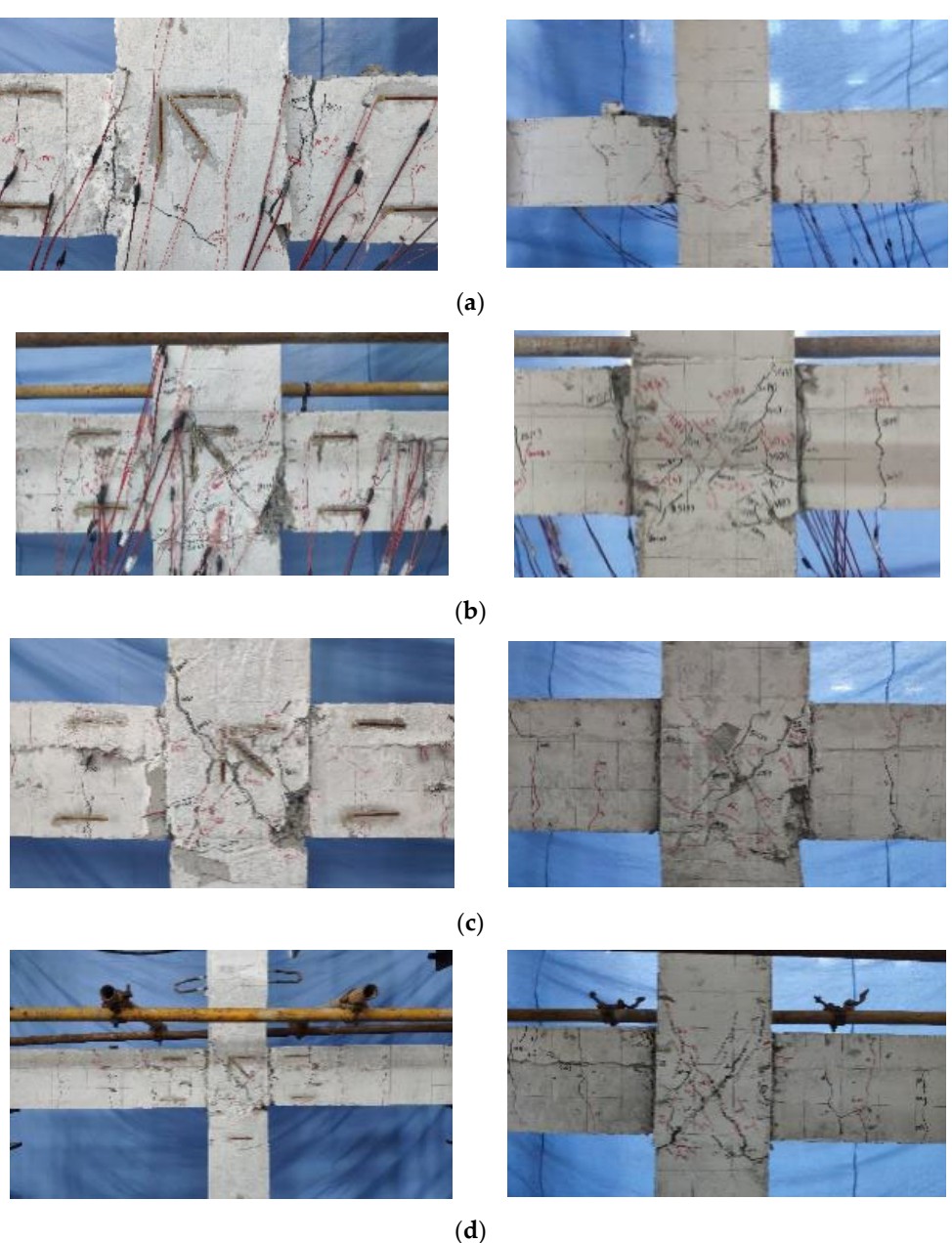

(a)

(b)

(c)

(d)

**Figure 6.** *Cont.*

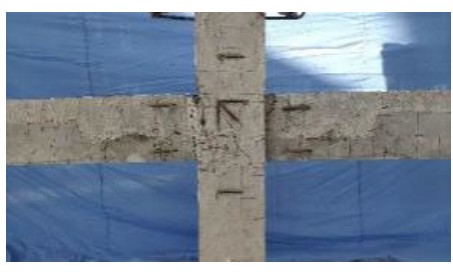
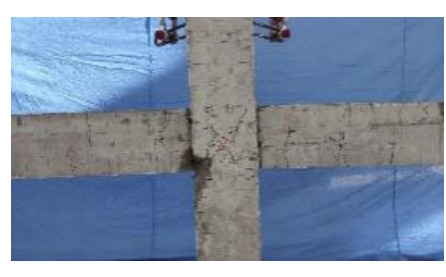

(**e**)

**Figure 6.** Failure modes of each specimen: (**a**) cast-in-site joint, (**b**) ZP-1, (**c**) ZP-2, (**d**) ZP-3, (**e**) ZP-4.

### 4.2. Interpretation of Hysteresis and Backbone Curves

Figure 7 illustrates hysteresis and backbone curves for each joint specimen, with the hysteresis curves for all five joint specimens resembling each other closely. The hysteresis loops displayed robust fullness and residual deformation post-unloading, typical of reinforced concretes. Despite the similar initial structures, the enduring loading phase revealed a pinching phenomenon. The bearing capacity of the ZP-2 joint decreased at a rapid pace, with recorded positive terminal values significantly lower than the other four specimens. The initial rigidity of ZP-2 was also markedly inferior to the other three new joints. The hysteresis loop area for ZP-2 was notably the smallest. Figure 7f reveals that the backbone curves for all joint specimens were "S" shaped, representing the elastic, plastic, limit, and failure stages. ZP-3's elastic phase curve initially displayed a steeper slope than the other four specimens but declined faster post-peak. Consequently, the ductility of ZP-3 was considered inferior due to diminished strength post concrete pouring. A comparison of ZP-4 with ZP-1 indicated minor differences in initial stiffness, with the pre-final-load slope of the curve being relatively consistent. However, ZP-4's final load surpassed that of ZP-1. Continuing to heighten the load led to a decline in joint bearing capacity, with ZP-1 declining at a notably faster rate than ZP-4. Compared to ZP-2, ZP-1 had superior bearing capacity, which decreased at a slower rate than ZP-2 post reaching the final load.

The improvement of bearing capacity, ductility, and energy dissipation capacity of joints can be effectively achieved by enhancing the post-casting area of concrete and the length of the L-shaped bars.

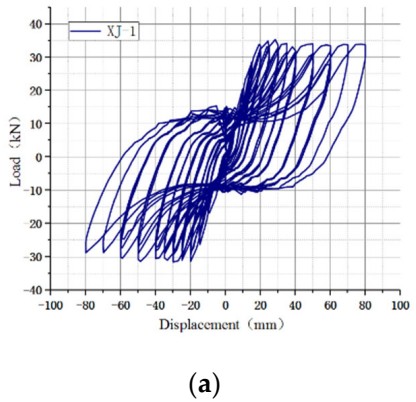
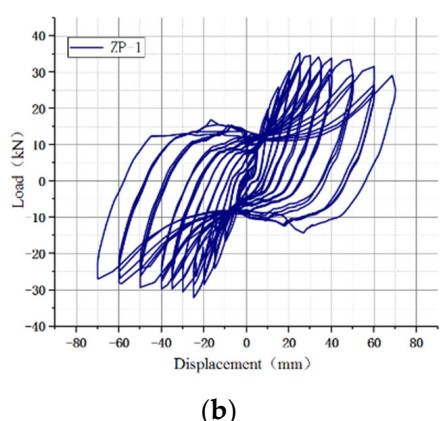

(**a**)  (**b**)

**Figure 7.** *Cont.*

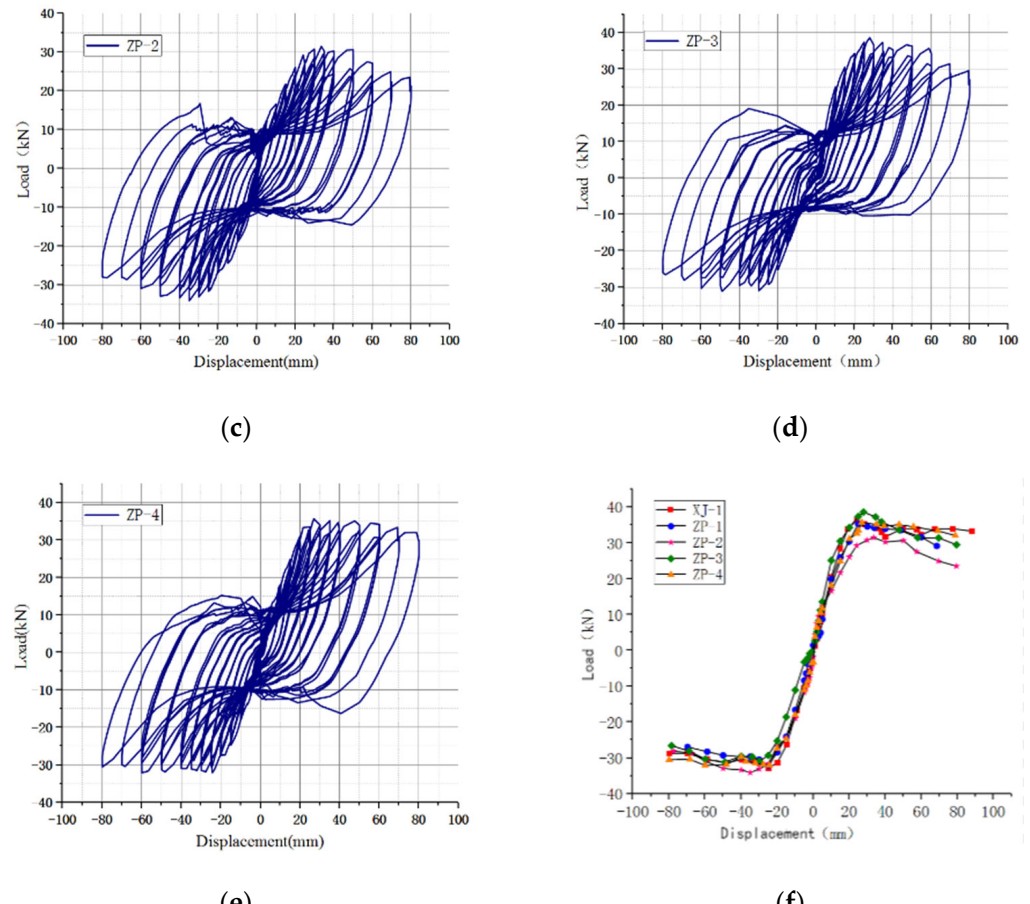

**Figure 7.** Hysteresis curve and skeleton curve: (**a**) XJ-1, (**b**) ZP-1, (**c**) ZP-2, (**d**) ZP-3, (**e**) ZP-4, (**f**) skeleton curve.

### 4.3. Analysis of Ductility

Ductility [26] reflects the deformation capacity of the joints following yielding. Its extent largely influences the seismic performance of the structure. It is typically represented by the ratio of the ultimate displacement $\Delta_u$ to the yield displacement $\Delta_y$. The equation for the ductility coefficient is as follows:

$$\mu = \frac{\Delta_u}{\Delta_y} \tag{1}$$

in which $\Delta_u$ represents the failure displacement of the joint specimen, specifically, the corresponding displacement when the load value diminishes to 85% of the peak load. $\Delta_y$ signifies the yield displacement of joint specimens. Both the load characteristic values and the ductility coefficients of each specimen are computed, and the findings are displayed in Table 4.

Based on the data in Table 4, the ductility coefficient of ZP-1 is marginally higher than that of ZP-2 and ZP-3, suggesting that an increase in the length of L-shaped bars can effectively enhance the ductility of joint specimens. However, a decrease in concrete strength in the composite beam area can lead to a decline in ductility. The ductility coefficient of ZP-4 is significantly larger than the other three newly assembled joints, indicating that ductility performance is amplified by augmenting the post-cast area of the concrete in the joint. The ductility performance of the cast-in-place joints is notably superior than the other four joints due to the integrity performance. The bonding of new joints between old and new concrete can influence the overall mechanical properties of the joints.

**Table 4.** Load characteristic values and ductility coefficient of specimens.

| Specimen Number | Loading Direction | Yield Point | | Ultimate Point | | Breakdown Point | | Ductility Coefficient | |
|---|---|---|---|---|---|---|---|---|---|
| | | $P_y$ (kN) | $\Delta_y$ (mm) | $P_m$ (kN) | $\Delta_m$ (mm) | $P_u$ (kN) | $\Delta_u$ (mm) | $\mu$ | |
| XJ-1 | positive | 33.25 | 18.68 | 35.29 | 28.42 | 33.92 | 77.58 | 4.15 | 4.38 |
| | reverse | −28.8 | −17.37 | −32.8 | −25 | −28.75 | −80 | 4.61 | |
| ZP-1 | positive | 31.39 | 20.96 | 35.25 | 24.88 | 29.96 | 65.58 | 3.12 | 3.22 |
| | reverse | −29.19 | −21.04 | −32.03 | −24.97 | −26.92 | −69.8 | 3.32 | |
| ZP-2 | positive | 26.60 | 20.67 | 31.45 | 29.86 | 26.73 | 60.96 | 2.95 | 3.05 |
| | reverse | −28.13 | −21.38 | −34.06 | −34.99 | −28.1 | −67.37 | 3.15 | |
| ZP-3 | positive | 33.05 | 18.40 | 38.57 | 27.85 | 32.78 | 51.55 | 2.8 | 2.87 |
| | reverse | −29.88 | −26.78 | −30.94 | −29.97 | −26.55 | −78.63 | 2.93 | |
| ZP-4 | positive | 31.53 | 21.18 | 35.66 | 26.89 | 30.31 | 78.75 | 3.72 | 3.96 |
| | reverse | −26.88 | −19.07 | −32.01 | −24.29 | −30.5 | −79.93 | 4.2 | |

Thus, it is observed that by enlarging the concrete post-cast area in the joint area, the length of the L-shape, and the concrete strength in the composite beam, the bonding capability between the post-cast area and the assembled component can be effectively realized with the goal of improving the ductility performance of the joints.

*4.4. Analysis of Energy Dissipation*

The energy dissipation capacity of the specimen under horizontal load is gauged using the area encompassed by the hysteretic loop in the hysteretic curve. The most commonly used assessment index is the equivalent viscous damping coefficient $h_e$, which is represented as follows:

$$h_e = \frac{1}{2\pi} \cdot \frac{S_{(EFGH)}}{S_{(\Delta OFM)} + S_{(\Delta OHN)}} \tag{2}$$

where $S_{(EFGH)}$ symbolizes the area enclosed by a hysteretic loop; $S_{(\Delta OFM)} + S_{(\Delta OHN)}$ symbolize the area enclosed by an imaginary elastic line at the same displacement. The visual explanation of $S_{(EFGH)}$ and $S_{(\Delta OFM)} + S_{(\Delta OHN)}$ is portrayed in Figure 8. The equivalent viscous damping coefficient $h_e$ of the five joints under the peak state is provided in Table 5.

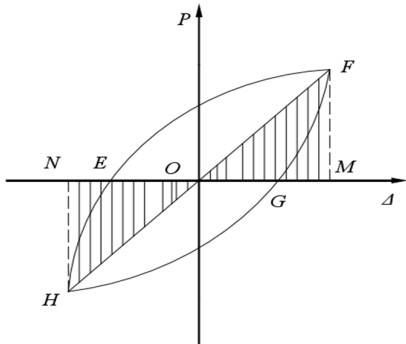

**Figure 8.** Calculation diagram of the equivalent viscous damping coefficient.

The analyzed data in Table 5 show that the equivalent viscous damping coefficient of the cast in situ joint XJ-1 is considerably higher than the rest of the four joints. The $h_e$ of ZP-2 and ZP-3 is notably smaller, suggesting a lower capacity to dissipate energy and more severe shear damage from the perspective of the failure process. It is also observable

that enhancing the length of L-bars, the concrete strength of the composite beam, and the post-cast area of concrete in the joint area can boost the energy dissipation capacity.

**Table 5.** Equivalent viscous damping coefficient.

| Joint Number | $S_{(EFGH)}$ (kN·mm) | $S_{(\Delta OFM)}+S_{(\Delta OHN)}$ (kN·mm) | $h_e$ |
|:---:|:---:|:---:|:---:|
| XJ-1 | 995.61 | 911.47 | 0.17 |
| ZP-1 | 696.83 | 839.81 | 0.13 |
| ZP-2 | 820.81 | 1065.43 | 0.12 |
| ZP-3 | 632.49 | 945.18 | 0.11 |
| ZP-4 | 890.53 | 919.56 | 0.15 |

*4.5. Analysis of Stiffness Degradation and Strength Degradation*

Under cyclic loads, when the same peak load is maintained, the phenomenon that the peak displacement increases with the increase of the number of cycles is called stiffness degradation [27]. Stiffness is measurable via secant stiffness, represented using the equation of secant stiffness, denoted as K:

$$K_i = \frac{|+P_i|+|-P_i|}{|+\Delta_i|+|-\Delta_i|} \tag{3}$$

In this equation, $+P_i$ is indicative of the load present at the forward peak point during the i-th cycle, while $-P_i$ the load at the negative peak point under the same cycle. $+\Delta_i$ and $-\Delta_i$ represent the forward-peak-point displacement and the negative-peak-point displacement at the i-th level of loading respectively. Post-calculation, the stiffness degradation curve can be visualized in Figure 9.

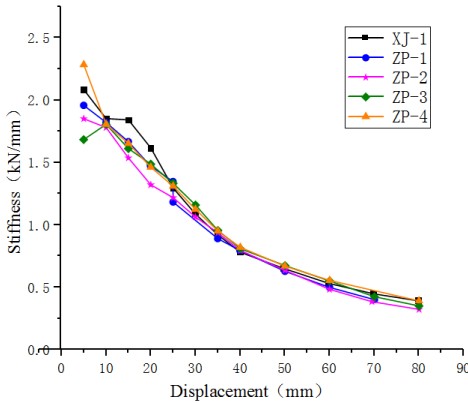

**Figure 9.** Stiffness degradation curve.

Our findings from five specimens were near-congruous in terms of overall stiffness and the degradation trends. The XJ-1 joint's complete stiffness smoothly transitions, with the initial stiffness of ZP-4 proving greater due to the inherent pipe, albeit with a high attenuation rate. ZP-2 demonstrates lower overall stiffness than the remaining joints, attributable to its shorter reinforcement length L, leading to shear failure in its core area. This outcome culminates in a lesser overall stiffness than that of the ZP-1 with longer L bars. Preservation of the steel pipe in the column can notably enhance the initial joint stiffness.

Under cyclic loading, with the increase of the number of load cycles at the same level, the strength of the specimen decreases continuously, which is called strength degradation [28], typically estimated by the strength degradation coefficient $\lambda_i$. The equation is as follows:

$$\lambda_i = \frac{P^i_{j,max}}{P^{i-1}_{j,max}} \tag{4}$$

where $P^i_{j,\,max}$ is indicative of the peak load of the hysteresis curve for the *i*-th cycle when the specimen load is at the j-th level. $P^{i-1}_{j,max}$ represents the hysteresis curve's peak load for the *i*-1 cycle when the specimen is under load at the *j*-th stage. The strength degradation curve is shown in Figure 10.

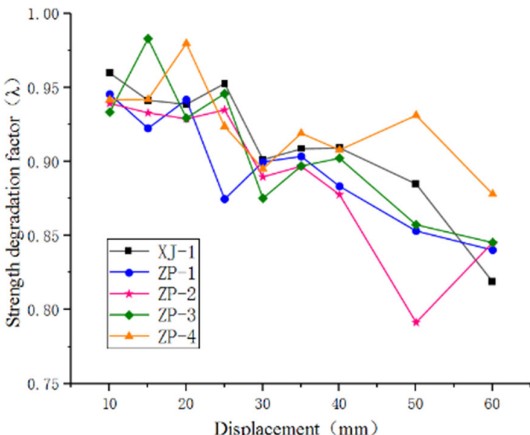

**Figure 10.** Strength degradation curve.

As displacement and load cycles amplify, the reinforcement of all specimens begins to yield. Periodically, characteristics such as the substantial fracturing of the concrete in the core area, serious damage in the plastic beam hinge area at the beam ends, and partial retreat from work lead to a decrease in joint strength. Additionally, due to the relative slip between concrete and longitudinal reinforcement, the joint strength proves irregular. Despite this, the strength of the five joints is stably maintained above 0.75. The strength degradation trend of cast-in-site joints and newly assembled joints show similar patterns, suggesting that the novel type of assembled joints possesses an excellent resistance to damage. However, the joint XJ-1 displays the most stable strength degradation. The new ZP-2 joint's strength experiences a sudden drop due to a large concrete wedge collapsing.

## 5. Finite Element Analysis and Shear Capacity Checking Calculation

To enable a comprehensive examination of the varying parameter influences on the performance of the newly assembled joints and to provide substantial data for the joint bearing capacity's theoretical validation, a finite element simulation and analyses of these joints were implemented based on the aforementioned tests.

### 5.1. Finite Element Modelling

Leveraging the capabilities of the finite element analysis software, namely ABAQUS 2020, we established four separate models of newly constructed concrete beam–column joints and a single model of a joint created in situ; these models were constructed in accordance with the test sizes.

The three-dimensional solid element responsible for concrete, C3D8R (linear reduced integration element), was utilized. The simulation and analysis of the concrete material was performed using the Concrete Damage Plasticity model (CDP model).

During the elastic stress phase of concrete, the mechanical aspects of the material were directly illustrated with the initial elastic modulus through the CDP model. However, upon the concrete reaching the material damage phase, the computational expression of the elastic modulus is modified as follows:

$$E = (1 - d)E_0 \tag{5}$$

where $E$ signifies the elastic modulus; $E_0$ represents the initial elastic modulus; and $d$ signifies the plastic damage factor, which varies from 0 to 1. Here, 0 implies there is no

material damage to the concrete, and 1 denotes complete damage, signifying no remaining strength.

The expression for the uniaxial stress–strain curve of concrete under tensile and compressive conditions is as follows:

$$\sigma_t = (1 - d_t)E_0\left(\varepsilon_t - \varepsilon_t^{pl}\right) \tag{6}$$

$$\sigma_c = (1 - d_c)E_0\left(\varepsilon_c - \varepsilon_c^{pl}\right) \tag{7}$$

where $\sigma_t$, $\sigma_c$ are the tensile and compressive stress, respectively; $\varepsilon_t$, $\varepsilon_t$ represent the tensile and compressive strain; $\varepsilon_t^{pl}$ and $\varepsilon_c^{pl}$ are the tensile plastic strain and compressive plastic strain of concrete, respectively; and $d_t$ and $d_c$ are the damage evolution coefficients for concrete under uniaxial tension and compression.

In the CDP model, the transformation of the yield surface or failure surface of the concrete material is chiefly regulated using the sum of two hardening variables. The stress–strain curves of concrete under uniaxial tension and compression along with the corresponding diagrams of cracking strain and inelastic strain are illustrated in Figures 11 and 12, respectively. Different damage factors describe the stiffness modification of concrete in uniaxial tension and compression. Refer to Table 6 for the concrete material parameters.

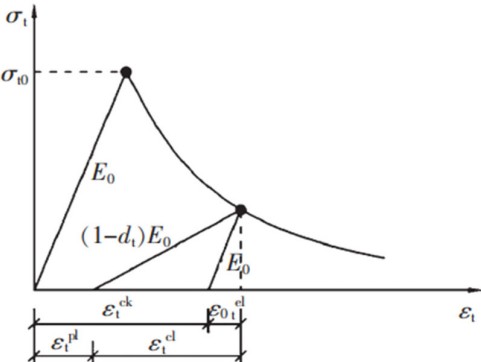

**Figure 11.** Uniaxial tension stress–strain curve and cracking strain diagram.

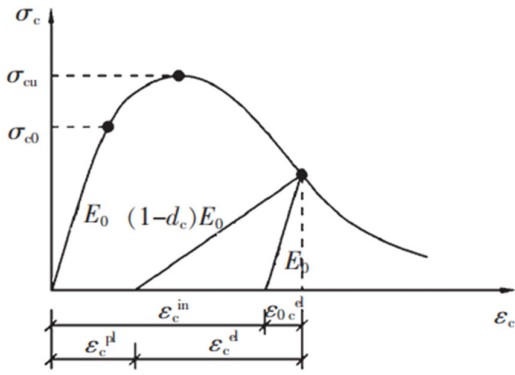

**Figure 12.** Uniaxial compression stress–strain curve and inelastic strain diagram.

**Table 6.** Concrete material parameter.

| Parameter | Poisson Ratio | Dilation Angle | Eccentricity Ratio | $f_{b0}/f_{c0}$ | K | Viscous Parameters | Compression Recovery Stiffness |
|---|---|---|---|---|---|---|---|
| Value | 0.2 | 38° | 0.1 | 1.16 | 2/3 | 0.01 | 0.6 |

In the model, the three-dimensional, two-node linear truss element T3D2, is utilized for reinforcement. The constitutive model for a steel bar, incorporated within, is a modified update of the double-fold steel bar constitutive model developed by Professor Fang Zihu of Shenzhen University. This model, as displayed in Figure 13, accounts for the bond-slip effect between the steel bar and concrete and considers the influence exerted by concrete materials on the steel bar. It is, therefore, an appropriate model to apply for the hysteresis analysis of the structure.

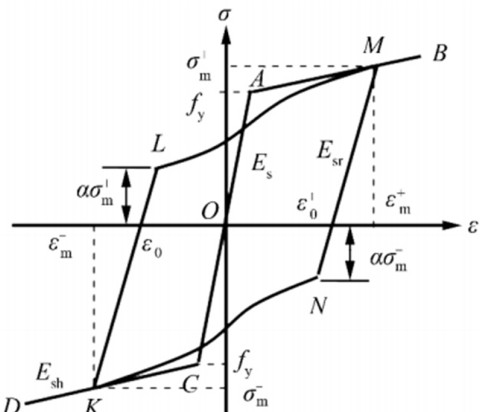

**Figure 13.** Modified constitutive model of steel bar.

Unloading stiffness calculation formula:

$$E_{sr} = \begin{cases} E_s, & \beta < 1 \\ (1.05 - 0.05\beta)\, E_s, & 1 \leq \beta \leq 4 \\ 0.85 \quad E_s, & \beta > 4 \end{cases} \tag{8}$$

Load path formula:

$$\sigma = \gamma\left(\bar{\varepsilon}^3 - \bar{\varepsilon}^2\right) + (1 - \alpha)\sigma_m\bar{\varepsilon} + \alpha\sigma_m \tag{9}$$

where $\gamma = E_{sh}(\varepsilon_m - \varepsilon_L) - (1 - \alpha)\sigma_m$; $\bar{\varepsilon} = (\varepsilon - \varepsilon_L)/(\varepsilon_m - \varepsilon_L)$; $\varepsilon_L$ is the strain corresponding to point $L$ or point $N$ in Figure 13; and $\alpha$ is the influence coefficient of hysteresis energy dissipation.

Finally, it is essential to take into account the elastic modulus, Poisson's ratio, yield strength, ultimate strength, and the associated parameters of reinforced concrete. These should be in accordance with the norms and measured values. Their specific values are exhibited in Table 7.

**Table 7.** Reinforcement of constitutive data.

| Steel Type | Diameter (mm) | Elasticity Modulus (N/m²) | Yield Strength Ultimate Strength (MPa) | Inelastic Strain |
|---|---|---|---|---|
| HRB400 | 6 | $2.0 \times 10^{11}$ | 429.32 | 0 |
| | | | 540.24 | 0.002 |
| HRB400 | 10 | $2.0 \times 10^{11}$ | 443.25 | 0 |
| | | | 625.16 | 0.002 |
| HRB400 | 12 | $2.0 \times 10^{11}$ | 450.13 | 0 |
| | | | 631.02 | 0.0021 |

In the FE model, three-dimensional solid elements are utilized for components such as the beam, column, and steel pipe. The composite beam's horizontal joint surface undergoes a process of artificial roughening during the construction phase, complemented with a sound stirrup configuration, allowing for a bond-slip-free horizontal composite surface. This surface would then necessitate the application of the "Tie" binding constraint. The interface relationship at the intersection of column joints and core areas is established by setting surface-to-surface contact. To designate contact attributes, one must separate the consideration of tangential behavior and normal behavior. For the tangential behavior, apply the "penalty" friction formula. Given that the surfaces of the upper and lower columns are irregular, the friction coefficient can be set at 1. Furthermore, normal behavior can be set to "hard contact," implying unrestricted force transmission in the normal direction. As for the vertical overlapping surface at the beam and column intersection point, an approximate "concrete softening" method is employed. This signifies that while the compressive strength of the concrete remains the same at the overlapping surface, the tensile strength will decrease to roughly 65%~85% of the original value. Compared to the horizontal laminate surface, the vertical one tends to bear not only shear stress but also more significant tensile stress. When it comes to the vertical composite joints of new and old concrete, compared to cast-in-place joints, they exhibit weaker bonding performance alongside lower tensile strength. However, their compressive strength mostly remains the same. Using the "concrete softening" method thus seems more feasible.

Use of the truss element for steel bars is common, and the standard mesh size stands at 50 mm. Figure 14 provides a representation of the joint models and mesh division. The beam–column connections' boundary conditions in the finite element simulation need to align with those established in the experiment.

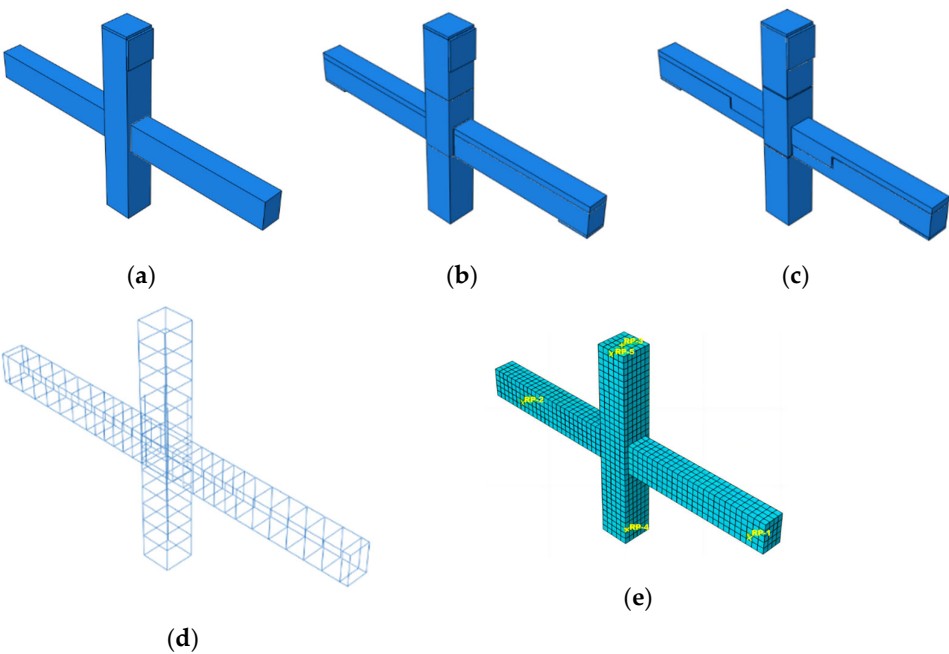

**Figure 14.** Modeling and meshing. (**a**) XJ-1, (**b**) ZP-1, ZP-2, ZP-3, (**c**) ZP-4, (**d**) reinforcing cage, (**e**) meshing.

### 5.2. Stress Cloud Diagram and Hysteresis Curve Analysis

The size of the preassembled unit cast in place uniformly resemble each other. The stress cloud for each joint model is available in Figure 15. The findings reveal that the longitudinal bars at the beam end tend to yield before the joint area's stirrups, which aligns with the real test results. The joint area's stirrups in the ZP-2 specimen yield before the longitudinal bars at the beam end, aligning yet again with the experimental results.

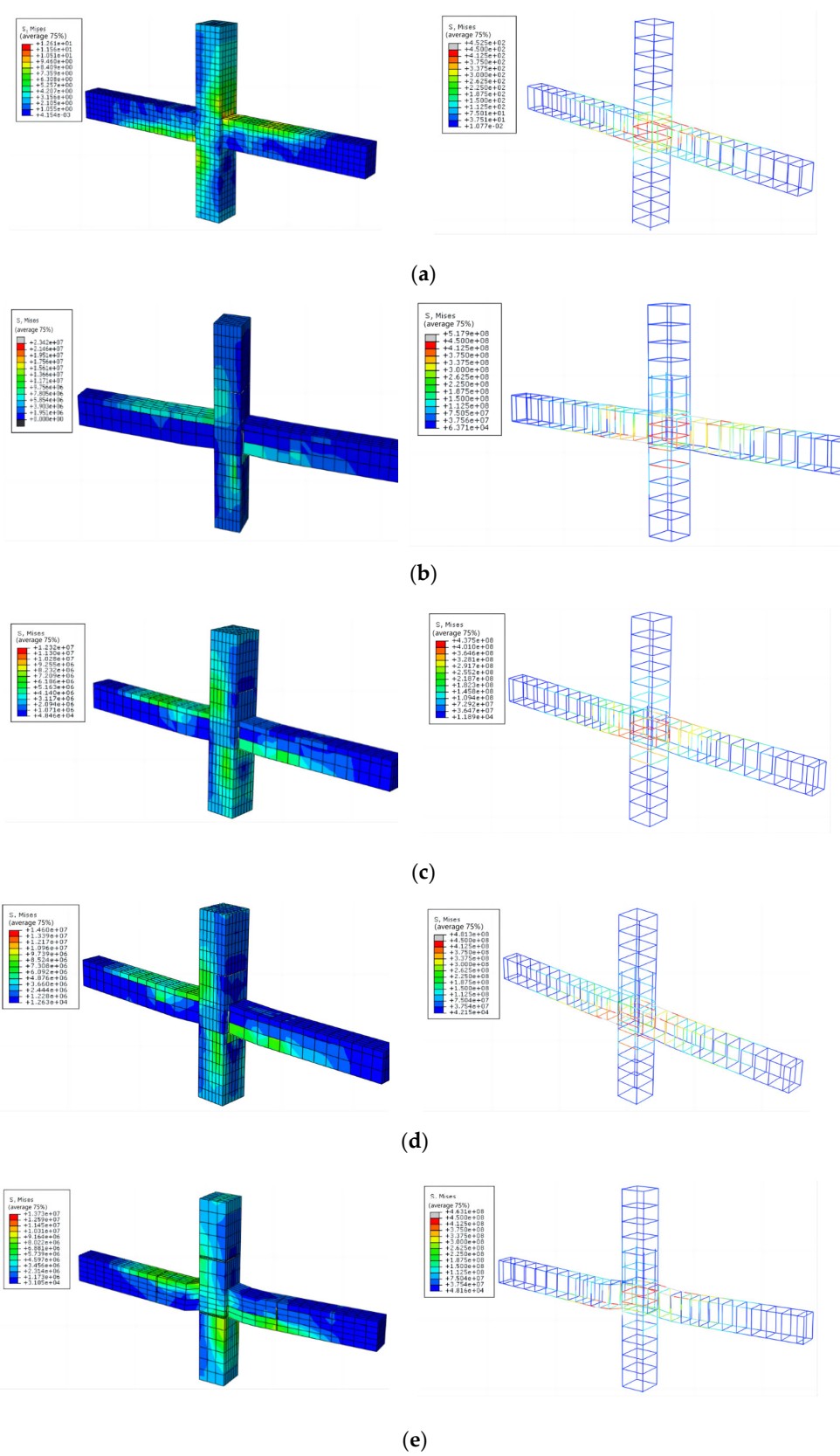

**Figure 15.** Stress cloud diagram. (**a**) XJ-1 joint, (**b**) ZP-1 joint, (**c**) ZP-2 joint, (**d**) ZP-3 joint, (**e**) ZP-4 joint.

Table 8 clearly shows that the overall hysteresis curve trend from the specimen, as shown in the finite element simulation, is substantially synonymous with the actual test. The peak points for each load also mostly align with the testing data.

**Table 8.** Hysteretic performance analysis of joints.

| Joint Number | Maximum Displacement (mm) | | Maximum Load (kN) | |
|---|---|---|---|---|
| | Test | Simulation | Test | Simulation |
| XJ-1 | 80.6903 | 101.2130 | 36.3741 | 31.8476 |
| ZP-1 | 29.2419 | 31.4079 | 35.7401 | 35.7401 |
| ZP-2 | 31.0588 | 32.4706 | 33.8583 | 35.5294 |
| ZP-3 | 31.7797 | 31.5254 | 38.8983 | 35.8475 |
| ZP-4 | 69.7161 | 80.1262 | 35.9528 | 35.9528 |

As shown in Figure 16, the finite element simulation hysteresis curve of the five nodal specimens is basically consistent with the overall change trend of the test hysteresis curve, and the peak points of each loading are basically consistent with the test data. Compared with the finite element simulation hysteresis curve, the test hysteresis curve has a large slip section in the later stage of loading, and the pinch phenomenon is obvious. This is because ABAQUS is difficult to fully simulate the bond slip between steel bars and concrete, so the hysteresis curve is relatively full. At the initial stage of loading, due to the gap between the loading device and the joint specimen, the displacement change speed is faster than the strain change speed. The finite element simulation is based on a simulation analysis under ideal conditions, so the initial stiffness of the finite element simulation hysteresis curve is significantly larger than that of the test hysteresis curve.

*5.3. Parameter Analysis and Shear Capacity Checking Calculation*

To investigate the impact of varying parameters on the seismic performance of joints, five new finite element models of assembled concrete with distinct axial compression ratios, post-cast concrete strength of joints, and steel tube diameters were developed. In these models, the strength grade of post-cast concrete is increased to C40, and the steel pipe specification is changed to an outer diameter of 60 mm and a wall thickness of 3.5 mm. These models are, respectively, designated as ZP-5.1, ZP-5.2, ZP-5.3, ZP-6, and ZP-7. Their specific parameters are articulated in Table 9.

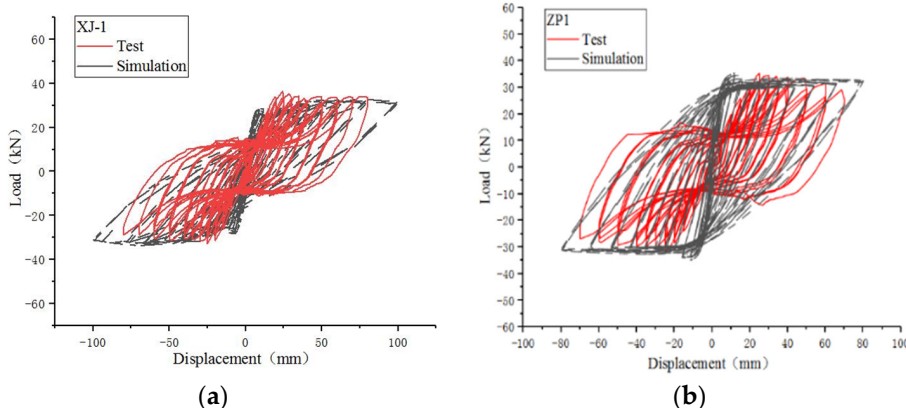

(a) (b)

**Figure 16.** *Cont.*

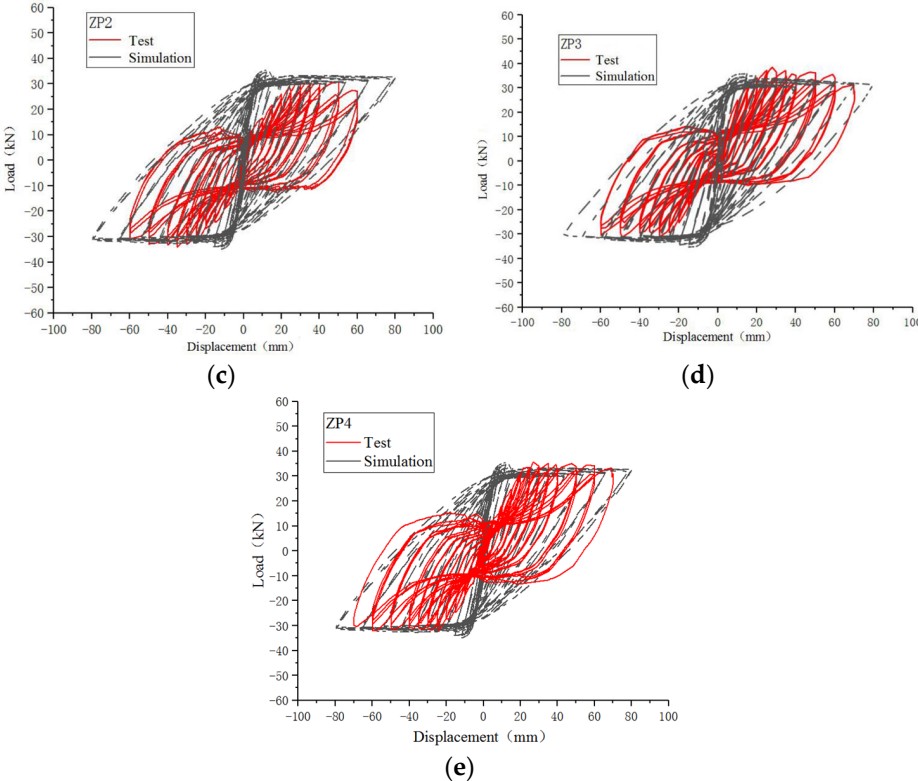

**Figure 16.** Hysteretic curve comparison between FEA and TEST: (**a**) XJ-1, (**b**) ZP-1, (**c**) ZP-2, (**d**) ZP-3, (**e**) ZP-4.

**Table 9.** Model parameters.

| Serial Number | Axial Compression Ratio | Strength of Concrete in the Joint Core Area | Steel Pipe Specification | |
|---|---|---|---|---|
| ZP-5.1 | 0.15 | C30 | D:42 mm | t:3 mm |
| ZP-5.2 | 0.25 | C30 | D:42 mm | t:3 mm |
| ZP-5.3 | 0.40 | C30 | D:42 mm | t:3 mm |
| ZP-6.0 | 0.20 | C40 | D:42 mm | t:3 mm |
| ZP-7.0 | 0.20 | C30 | D:60 mm | t:3.5 mm |

where D is outside diameter; t is wall thickness.

Investigations were conducted into the effects of different axial compression ratios and the increase of concrete strength in the joint area on the initial stiffness, ultimate load-bearing capacity, and ductility of the joint specimens. The findings are represented through a skeleton curve as depicted in Figure 17.

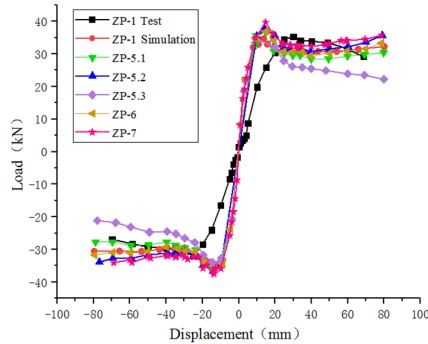

**Figure 17.** Skeleton curve.

Analyses of the backbone curves in Figure 17 show that the backbone curves of ZP-5.1, ZP-5.2, ZP-5.3, and ZP-6 are fundamentally akin, each featuring an "S" shape. The initial stiffness offered from the numerical simulation results is noticeably higher than those yielded from the test results. Moreover, the overall bearing capacity observed in the ZP-5.1 joint is marginally superior to that from the ZP-1 simulation. Conversely, the ultimate load capacity from the ZP-5.2 joint is almost identical to the ZP-1 simulation, albeit with the former declining at a faster rate and demonstrating poor ductility. The initial stiffness and ultimate load capacity of the ZP-4 joint are in line with the simulated results of the ZP-1; however, the load-bearing potency of the ZP-4 joint experiences a significant decline after achieving peak load, which is considerably lower than the simulated results of ZP-1 and has limited ductility. After increasing the concrete strength of the ZP-6 joint to C40, the initial stiffness, ultimate load-bearing capacity, and ductility essentially match those of the ZP-1 simulation results. The results for ZP-7 show improved initial stiffness, overall load-bearing capacity, and ductility.

The analysis suggests that, within a specific range, axial force is beneficial for shear resistance, aiding the advancement of initial stiffness and ultimate load-bearing capacity. However, when the axial force surpasses a certain limitation, it diminishes the joint's ductility. Enhancements to initial stiffness, ultimate load-bearing capacity, and ductility properties are achieved by augmenting the strength grade of the concrete in the joint's post-pouring region and incorporating steel piping into the column.

As per the Chinese GB50010 Concrete Structures Design Code (hereinafter referred to as "Concrete code") [29], the shear capacity of reinforced concrete joints is determined by the joint core concrete and stirrups. Nevertheless, introducing an in-built steel pipe in the new type of joint significantly enhances the shear load-bearing capacity. Consequently, the calculation of the new connection's shear capacity is proposed as follows:

$$V_j \leq \frac{1}{\gamma_{RE}} \left[ 1.1 \eta_j f_t b_j h_j + 0.05 \eta_j N \frac{b_j}{b_c} + f_{yv} A_{svj} \frac{h_{b0} - a'_s}{s} + f_v A_v \right] \tag{10}$$

where $\gamma_{RE}$ is 0.85 when calculating the inclined sectional bearing capacity of frame joints; $\eta_j$ is 1.50 for on-site cast floor slabs, but the recommended distribution, however, for a 9-degree fortification intensity is 1.25. For other conditions, the value is 1.00; $A_{svj}$ refers to the total cross-sectional area of each leg of the stirrup within the effective checking width range of the core area in the same cross-sectional direction; $f_{yv}$ is the design value of tensile strength for transverse reinforcement; $a'_s$ is the distance between the resultant point of reinforcement in the longitudinal compression of the beam and the section's near edge; $b_j$, $h_j$ are the effective check width and height of the core area's cross-section for frame joints; $h_{b0}$, $b_c$ are the effective height and width for the beam section; and $N$ is the design value of the axial force corresponding to the bottom of the upper column of the joint. Considering the seismic combined shear force's design value, when in compression, take the lower design value of axial pressure, and it should be less than $0.5 f_c b_c h_c$. When in tension, the value is zero.

Based on test data in Table 3, the joint core area's shear force value can be obtained using Equation (10) from the concrete code [29], as shown in Table 10.

$$V_t = \frac{\sum M_b}{h'_b} \left( 1 - \frac{H_b}{H_c - h_b} \right) \tag{11}$$

where $h_b$ is the beam section; $h_b'$ is the core area height of the joint; $\Sigma M_b$ is the sum of bending moments at the left and right beam ends of the joint; $H_b$ is the distance between the bending points of the left and right beams; and $H_c$ is the distance between the bending points of the upper and lower columns.

**Table 10.** Checking the calculation of shear bearing capacity.

| The Joint Type | Shearing Capacity (kN) | | | | |
| --- | --- | --- | --- | --- | --- |
| | Test | Theory | Error Value | Simulation | Error Value |
| Cast-in-site joint | 229.02 | 211.34 | 8% | 246.9 | 8% |
| ZP-1 | 220.29 | 239.41 | 7% | 250.66 | 14% |
| ZP-2 | 211.74 | 239.41 | 11% | 230.33 | 9% |
| ZP-3 | 226.21 | 239.41 | 5% | 249.88 | 10% |
| ZP-4 | 240.11 | 239.41 | 1% | 252.55 | 5% |

A comparison and analyses of the shear capacity from a theoretical calculation, experimental study, and finite element simulation show that the results in Table 10 are relatively similar. The error in shear capacity results is less than 15%, indicating that the shear capacity equation proposed by the newly assembled joint is reasonable and meets design and application requirements.

According to the study, to maximize the seismic performance of the newly assembly joints, the following requirements should be met. The experimental results show that L-shaped reinforced fabricated joints perform well, and performance improves with increased pouring area; therefore, it is recommended that precast beam ends are L-shaped. As per Chinese specifications, the value should not be less than 15 d to ensure performance. In design, higher concrete strength in the back pouring area leads to better joint performance. Therefore, it is recommended to use high-strength concrete for prefabrication, and the strength grade should be one level higher than the frame beams and columns.

## 6. Conclusions

This study explores the limitations of conventional integrated joint assemblies and introduces a fresh, assembled concrete beam–column L-shaped reinforced joint model. The research encompasses the conceptualization and production of one cast-in-place joint and four assembled joint exemplars, as well as low-cycle cyclic loading tests to investigate the mechanical mechanisms, failure modes, and seismic performance of these joints. A nonlinear analysis of the joints is executed using ABAQUS, with simulation and test results compared to each other. The model's accuracy is verified, and a parametric analysis is conducted, resulting in a formula for calculating shear capacity. Key conclusions include:

1. The size reinforcement of the beam column itself is designed according to the specification, and the column steel bar is continuous when connected, and the beam steel bar is bent to the column. Therefore, the design and production of the beam column itself has met the requirements of the strong column and the weak beam. Cast-in-place joint XJ-1 and newly assembled joints ZP-1, ZP-3, and ZP-4 exhibit bending failures at beam ends, complying with the seismic design requirements of "strong joints and weak members." Joint ZP-2 uses short L-shaped reinforcement, effectively reducing the vertical force component while increasing the concrete load, leading to shear failure in the joint area.

2. The newly assembled joints demonstrate excellent energy dissipation, damage resistance, and ductility. The stiffness degradation coefficient is above 0.75, indicating impressive damage resistance within the cast-in-place joints.

3. The force at the joint is more complex, and there are many factors affecting the seismic performance of the joint. In this paper, the influence of the main factors on the seismic performance of joint specimens has been studied. The study reveals that implementing measures such as increasing the area of post-cast concrete in the joint, extending the length of L-shaped reinforcement, and enhancing the concrete strength in the composite beam area can effectively improve the bonding force between the post-cast and precast members, subsequently boosting the joint's ductility, energy dissipation, and load-bearing capacity. Pre-placing steel tubes in columns can significantly enhance the initial stiffness of joints. It provides an effective basis for the research of the

seismic performance of the joint in the future, and promotes the popularization and application of the newly assembled joint.

4.  A comparative analysis confirms the strong alignment of the theoretical calculations with experimentally and numerically derived shear capacity values. The results show that the proposed formula meets the needs of design and application, and provides a reliable reference for the calculation and analysis of the shear capacity of the newly assembled joints.

Beam–column joints represent a pivotal research focus for the seismic analysis of assembled structures. The paper proposes a novel assembled concrete beam–column L-shaped reinforced joint and preliminarily investigates its seismic performance via pseudo-static testing and finite element simulation, but there are still shortcomings. For the successful development and application of this new joint model, the following considerations should be prioritized:

1.  Although newly assembled joints show promising feasibility for the assembled building industry by satisfying requirements for a simple structure, ease of construction, and structural integrity, their performance is influenced by various factors, which warrant close attention during practical applications, such as L-shaped reinforcement length, post-cast concrete area in the joints, and composite beam area concrete strength. In this paper, the effects of various factors such as the length of the L-shaped steel bar, the area of concrete poured after the joint, and the strength of the concrete in the composite beam area are studied through experiments. However, changing the axial compression ratio and increasing the strength of concrete in the joint area are realized through finite element simulation, and the simulation results are still different from the actual stress characteristics. Subsequent researchers can add tests to verify these results. At the same time, the effect of reinforcement ratio of the horizontal stirrups, vertical reinforcement, and column longitudinal reinforcement on the seismic performance of the joint should be further considered.

2.  The test specimens included in this study utilized a 1/2 scaled-down model to abide by actual testing equipment and site constraints. Assessing the seismic performance of four assembled joint specimens provided preliminary insights. However, due to the complex forces at the joints, it is often difficult to simulate the actual structural effect of full-size specimens after reducing the size, so subsequent researchers can make a small number of full-size specimens based on this scaling experiment to supplement and verify the results.

3.  Owing to the minor discrepancies between the finite element simulation outcomes and test results, we have endeavored to rectify these deviations. The ultimate finite element simulation findings show excellent agreement with the test outcomes, effectively simulating the maximum displacement and maximum load-bearing capacity. This considerably advances the scope of the research. Nevertheless, the issue of concrete material closure in finite element simulations remains unresolved, representing a longstanding challenge in the field of finite element analysis. In-depth investigations are necessary for future development and progress.

4.  In this study, the existing design formula for preassembled joints outlined in the Chinese Code is referenced, and we put forth a novel design formula to determine joint shear capacities. Comparing the theoretical calculations and analyses outcomes against experimental and finite element simulation results reveals a strong correlation among them. Although the proposed load-bearing capacity formula offers significant reference value and practical applicability to some extent, its use in other joint configurations with varying parameters warrants further examination using alternative formulae.

In summary, the innovative assembled L-shaped steel bar connection joints demonstrate practical feasibility, and initial research findings have been achieved. By employing analytical test and simulation methods, the assembled joint performance was assessed, verifying that they meet the necessary requirements and hold promising development potential.

**Author Contributions:** Conceptualization, M.L. (Mengjiao Lv) and T.Y. and M.L. (Mingqiang Lin); methodology, M.L. (Mengjiao Lv) and T.Y.; software: M.L. (Mengjiao Lv) and T.Y.; validation, M.L. (Mengjiao Lv) and T.Y.; formal analysis, M.L. (Mengjiao Lv) and T.Y.; investigation, M.L. (Mengjiao Lv) and T.Y.; resources, M.L. (Mengjiao Lv) and T.Y.; data curation, M.L. (Mengjiao Lv) and T.Y. and M.L. (Mingqiang Lin); writing—original draft, M.L. (Mengjiao Lv) and T.Y.; funding acquisition, T.Y. All authors have read and agreed to the published version of the manuscript.

**Funding:** This research was funded by [Natural Science Foundation of Shandong Province] grant number [ZR202102180176] and [Municipal and School Integration Development Strategic Project of Jinan City] grant number [JNSX2023023].

**Data Availability Statement:** The data presented in this study are available on request from the corresponding author. The dates are not publicly available due to the privacy of the data.

**Conflicts of Interest:** The authors declare no conflicts of interest.

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
