# Peer review of "Experimental Study on Seismic Behavior of Newly Assembled Concrete Beam–Column Joints with L-Shaped Steel Bars"

_applsci, doi:10.3390/app14031262_

Round 1

Reviewer 1 Report

Comments and Suggestions for Authors

Detailed comments for the authors can be found inside the attached PDF file.

Comments on the Quality of English Language

Minor improvements to English language are needed.

Reviewer 2 Report

Comments and Suggestions for Authors

Dear Authors, 

Suggested corrections are listed below;

Please add 1-2 sentences with the numerical results obtained in the abstract section. The abstract will be the most read part of the study.

In the study, it is recommended to first give information about the post-earthquake damages in the column-beam joints. In this way, the importance of the subject will be more understandable. Some suggested and similar references could be used. Suggested references;

Ozturk et al (2023). Effect on RC buildings of 6 February 2023 Turkey earthquake doublets and new doctrines for seismic design. Engineering Failure Analysis153, 107521.

Caglar et al (2023). Structural damages observed in buildings after the January 24, 2020 Elazığ-Sivrice earthquake in Türkiye. Case Studies in Construction Materials18, e01886.

Isik et al. (2020). 24 January 2020 Sivrice (Elazig) earthquake damages and determination of earthquake parameters in the region. Earthquakes and Structures19(2), 145.

It would be beneficial to expand the literature section in order to more clearly reveal the novelty or difference of the study. It is useful to choose the current ones from different and similar studies proposed for strengthening the column-beam areas.

You can add the stages of your work to the last paragraph of the introduction. At the end of this paragraph, you should clearly state the novelty/difference of your work.

The limitation of the study should be specified and the situations in which it would be more effective should be discussed in detail.

Please use the same text format for all figures.

Authors should add comments about weak beam - strong column controls after strengthening.

If possible, it is recommended to compare the results of different strengthening in these regions.

It is recommended to increase the resolution of some figures.

In the conclusion part, please include the contribution of your work to practice and to such studies in the future.

Reviewer 3 Report

Comments and Suggestions for Authors

The seismic performance of beam-column joints in prefabricated structures with a proposed steel reinforcement detailing is investigated experimentally and numerically in this manuscript. The experimental program and results presented in the manuscript are well-presented and enrich the literature. However, this reviewer has some concerns related to the FEA results and recommends to revise the manuscript to improve it by addressing the following comments carefully:

- The comparisons presented in Fig. 16 show poor correlation between the FEA and test results, particularly regarding the initial stiffness, strength degradation, and pinching effect. The authors should work on improving the FEA models and revising these results or provide solid justification for the differences between FEA and test results.

- Some of the expressions used in the manuscript should be revised, such as the following:

-        Lines 15-16: Replace “nodes” with “joint or connection”

-        Line 17: Replace “skeleton curves” with “backbone curves”

-        Line 248: Replace “Energy Consumption” with “Energy Dissipation”

- Some of the manuscript sentences are difficult to read, and some typos and grammatical mistakes are noted. It is highly recommended that the paper be proofread to improve its readability and correct typographical mistakes. For instance, in Lines 20-22, rephrase the following sentence for clarity “The hysteresis curve of the newly developed prefabricated joints is relatively full…”. Also, rephrase the definitions of stiffness/strength degradation in Lines 269-271 and Lines 288-290 for clarity and provide additional supporting references.

Comments on the Quality of English Language

Some of the manuscript sentences are difficult to read, and some typos and grammatical mistakes are noted. It is recommended that the paper be proofread to improve its readability and correct typographical mistakes. 

Round 2

Reviewer 3 Report

Comments and Suggestions for Authors

The Authors have addressed the Reviewer’s comments, and, in my opinion, the manuscript is worth publishing now

Comments on the Quality of English Language

Minor editing of English language may be needed